# A Step to Develop Heat-Health Action Plan: Assessing Heat Waves' Impacts on Mortality

Hazal Cansu Çulpan [1,*] , Ümit Şahin [2] and Günay Can [3]

1   Karaman Central Community Health Center, Gevher Hatun, Karaman 70200, Turkey
2   Sabanci University Istanbul Policy Center, Bankalar Caddesi, No:2 Karaköy Minerva Han, Istanbul 34420, Turkey
3   Department of Public Health, Cerrahpasa Faculty of Medicine, Istanbul University-Cerrahpasa, Fatih, Istanbul 34098, Turkey
*   Correspondence: hazal.acar@istanbul.edu.tr or hazalcansu@gmail.com

**Abstract:** Climate change is one of the biggest health threats facing humanity and can directly affect human health through heat waves. This study aims to evaluate excess deaths during heat waves between the summer months of 2004 and 2017 in Istanbul and to determine a definition of heat waves that can be used in the development of an early warning system, a part of prospective urban heat-health action plans. In this study, heat waves were determined using the Excess Heat Factor, an index based on a three-day-averaged daily mean temperature. The death rates during heat waves and non-heat wave days of the summer months were compared with a Z test of the difference of natural logarithms. Thirty heat waves were recorded in Istanbul during the summer months of 2004–2017. In 67% of the heat waves, the death rate was significantly higher than the reference period and 4281 excess deaths were recorded. The mortality risk was especially higher during heat waves of higher intensity. The study showed an excess risk of mortality during heat waves in Istanbul, and the findings suggest that the Excess Heat Factor could be an appropriate tool for an early warning system in Istanbul.

**Keywords:** heat wave; excess heat factor; Istanbul; excess death; heat-health action plan; early warning system; extreme weather and climate events

## 1. Introduction

Anthropogenic climate change causes many extreme weather events all over the world. There is growing evidence of human influence on changes in extreme weather events such as heat waves, heavy rain, and droughts [1]. Studies show that not only the frequency, but also the duration and intensity of heat waves—one of the most common direct effects of anthropogenic climate change—have increased [2,3]. According to the summary of the Sixth Assessment Report of the Intergovernmental Panel on Climate Change (IPCC) published in August 2021, it is almost certain that extreme heat waves have become more frequent and more intense since the 1950s [1]. In the Eastern Mediterranean region, where Turkey is located, an increase in heat waves has been observed since the 1960s [4].

The negative impact of excess heat on human health has been documented for over a century [5–9]. A study conducted in the city of St. Louis, Missouri, United States, between 1910 and 1912 shows the association between the mortality of infants suffering from gastrointestinal disorders and heat peaks [5]. Over 20% more deaths were reported in the 1976 heat wave in the British Isles, for example [6]. The 2003 heat wave in Europe caused more than 70,000 excess deaths in a total of 16 countries [7]. In the 2010 heat wave in India, more than 1300 excess deaths were reported in Ahmedabad alone [8]. Extreme heat waves in 2017 in Pakistan caused one thousand excess deaths [9].

Due to its location in the Mediterranean Basin and the desert climate expanding to the north, Turkey is one of the countries that will be the most affected by climate change [10,11].

Current studies and the projections performed with global climate models show that there are more frequent heat waves in Turkey than in the past, and that the country will soon be hotter and drier than it is now [10–12]. It has been reported that the number of hot days and the frequency, duration, and intensity of heat waves have increased since the 1960s in the western part of Turkey, including Istanbul, the most populous city in Turkey with a population of approximately 15.5 million in 2020 [4,13,14]. Hence, a large number of people are at risk of experiencing the negative health effects of hot weather in Istanbul.

Reducing the negative health effects of excess heat on the public is possible through conducting further scientific research, establishing local and national preparedness measures, and implementing public health policies. In order to prevent these effects, the World Health Organization (WHO) recommends the implementation of Heat-Health Action Plans, which include taking actions at different levels of society such as the creation of an early warning system, preparedness in the health system, and improvements in urban planning [15].

Although there are a limited number of studies on the relationship between heat waves and health outcomes in Turkey, recent studies illustrate the negative impact of heat waves on health [16–19]. Therefore, there is a need for local and national action plans for Turkey. In order to prepare effective action plans and develop early warning systems, first, a standard definition of heat wave should be outlined at the local level and the current impacts of heat waves should be determined through scientific research.

This study aims to assess excess deaths during heat waves between 2004 and 2017 in Istanbul and to establish a definition of heat waves that could be used in the development of an early warning system, a part of Istanbul's future heat-health action plans.

## 2. Materials and Methods

### 2.1. Study Area

This health impact study was carried out in Istanbul, which is located between and along the Black Sea and the Sea of Marmara. The southern parts of Istanbul have a Mediterranean climate, in which summer is generally hot and dry and winter is warm [20]. The northern parts of the city are cooler in both summer and winter due to the effects of the Black Sea and receive more precipitation than the south [21].

Istanbul is the 15th most populated city in the world according to the 2018 Revision of the United Nations World Urbanization Prospects [22]. However, the population density in the city varies greatly between districts [23]. The population density map of Istanbul is given in Figure 1.

### 2.2. Meteorological Data

The meteorological data from Istanbul between the years 1971 and 2017 was provided by the First Regional Directorate of the Turkish State Meteorological Service from four meteorological stations in Istanbul (Figure 1).

The daily mean temperature ($T_i$) was calculated by using the daily minimum and maximum temperatures. However, since the physiological response to a hot night following a hot day is more significant than the vice versa, when calculating $T_i$ of any day, the maximum temperature of the day and the minimum temperature of the next day, in other words, the following night, were averaged [24].

### 2.3. Mortality Data and Ethics Approval

The data on the daily number of deaths in the summer months (June–August) between the years 2004 and 2017 were obtained from the Istanbul Metropolitan Municipality Department of Cemeteries. All-cause mortality was included in the study. Crude death rates during heat waves and during the reference period were calculated. The population data used in the calculations were obtained from the Turkish Statistical Institute [14].

The study was approved by the Clinical Research Ethics Committee of Istanbul University-Cerrahpasa (approval number: 43633, approval date: 3 March 2021). The

requirement for written informed consent was waived because of the retrospective nature of the study and the data were anonymized before being obtained for the study. The study was reported according to the Strengthening the Reporting of Observational Studies in Epidemiology (STROBE) Statement.

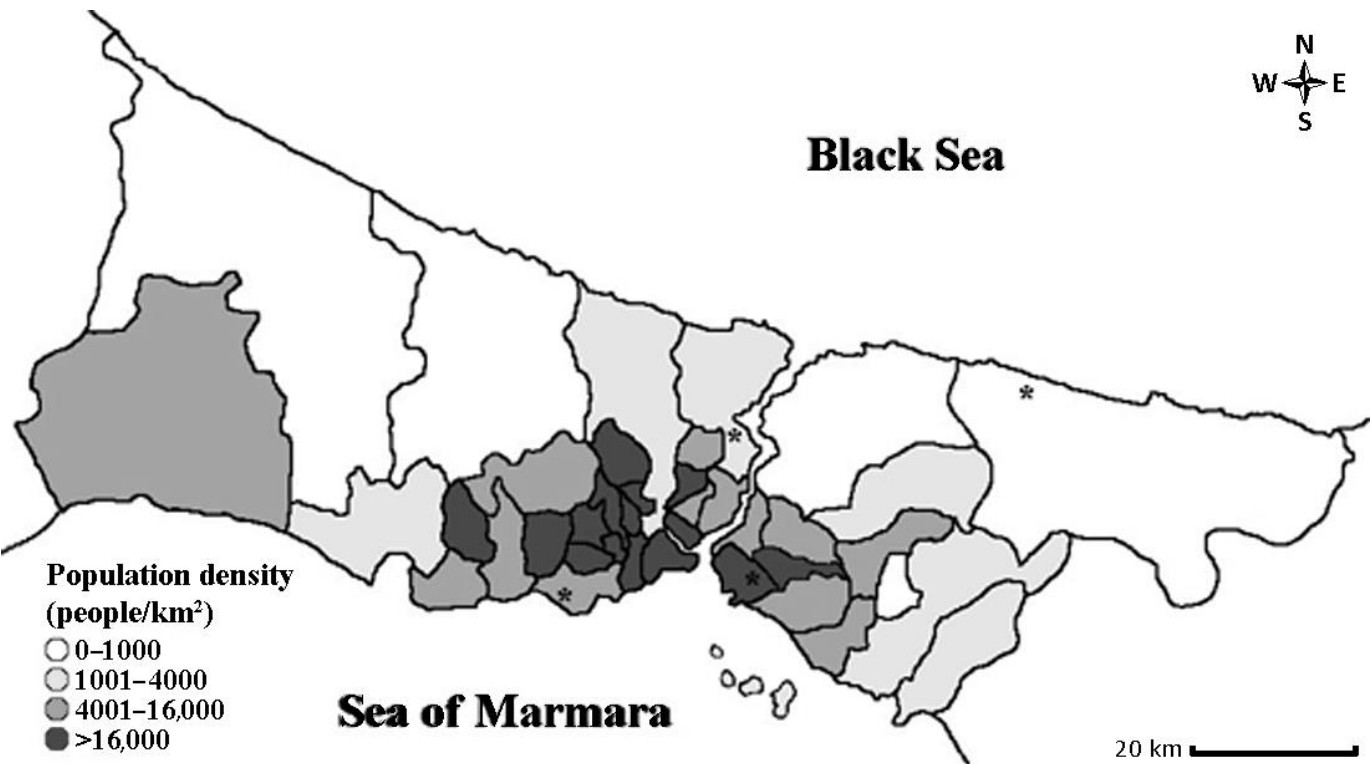

**Figure 1.** Population density (people/km$^2$) map of Istanbul by districts. Asterisk (*) shows location of meteorological stations. Source note: population data obtained from Istanbul Statistics Office of Istanbul Metropolitan Municipality [23].

*2.4. Definition of Heat Wave and Reference Period*

The Excess Heat Factor (EHF), a recently introduced metric by the Australian Bureau of Meteorology, was used to identify heat waves and their intensity [24]. EHF is also one of the indicators included in the guidelines for establishing an early warning system published by the WHO and World Meteorological Organization (WMO) [25]. It consists of two indices called the Excess Heat Indices (EHIs). The first index is the significance index (EHI$_{sig}$), which measures the temperature for a consecutive three-day period (TDP) in a given location relative to the historical temperature threshold. It evaluates whether the average T$_i$ for a TDP is higher than the 95th percentile (T$_{95}$) of historical temperatures, which are usually based on a 30-year reference period.

The second index is the acclimatization index (EHI$_{accl}$). It measures the difference in the average T$_i$ between a TDP and the 30 days before it. In other words, it measures the short-term acclimatization of people. EHI$_{sig}$ and EHI$_{accl}$ were calculated as Equations (1) and (2), respectively.

$$EHI_{sig} = \frac{(T_i + T_{i+1} + T_{i+2})}{3} - T_{95} \tag{1}$$

$$EHI_{accl} = \frac{(T_i + T_{i+1} + T_{i+2})}{3} - \frac{(T_{i-1} + \cdots + T_{i-30})}{30} \tag{2}$$

A positive EHI$_{sig}$ and/or EHI$_{accl}$ indicates a temperature anomaly. In this case, EHF was calculated as Equation (3).

$$EHF = EHI_{sig} \times maks(1, EHI_{accl}) \tag{3}$$

If three consecutive EHF are positive, then all days within the TDP are considered to be heat wave days, and together they form a heat wave of at least 5 days duration. Low EHF values indicate that the intensity of the heat wave is low, while high EHF values indicate higher intensity. The intensity of heat waves was assessed using the maximum EHF and total EHF (sum of all EHF values) during a heat wave [26]. The 25th, 50th, 75th, and 95th percentiles of the EHF were calculated using the EHF values of all heat waves [27]. Those percentiles were used for identifying the intensity pattern of heat waves.

In the study, $T_i$ for the summer months from 1971–2017 were calculated. The $T_i$ between 1971 and 2000 was used to determine the $T_{95}$ of June, July, and August. Then, the EHF was calculated using the above formulas to identify heat waves in the summer months of 2004–2017. A reference period of a heat wave was defined as non-heat wave days in summer months of the same year. The mortality rate of a heat wave was compared to the mortality rate of the reference period.

*2.5. Statistical Analysis*

Statistical analyses were performed using SPSS v21.0 (SPSS Inc., Chicago, IL, USA) and Microsoft Excel (Microsoft Corporation, Redmond, WA, USA). The crude death rate during a heat wave ($CDR_{HW}$) was calculated by dividing the number of deaths ($m_{HW}$) by the population (P) multiplied by the length of the heat wave ($d_{HW}$). Likewise, the crude death rate during the reference period ($CDR_{Ref}$) was calculated by dividing the number of deaths ($m_{Ref}$) by the population (P) multiplied by the length of non-heat wave days ($d_{Ref}$). The $CDR_{HW}$ and $CDR_{Ref}$ were calculated with Equations (4) and (5), respectively.

$$CDR_{HW} = \frac{m_{HW}}{P \times d_{HW}} \times 100000 \tag{4}$$

$$CDR_{Ref} = \frac{m_{Ref}}{P \times d_{Ref}} \times 100000 \tag{5}$$

Death rates during a heat wave and the reference period were compared using Z tests of the difference of natural logarithms [28].

The rate ratio (RR) was calculated by dividing $CDR_{HW}$ by $CDR_{Ref}$. The results were presented with 95% confidence intervals (95% CI). The excess deaths during heat waves were calculated by multiplying the difference of $CDR_{HW}$ with $CDR_{Ref}$ by the population and $d_{HW}$. Statistical significance was accepted as $p < 0.05$.

**3. Results**

The average number of deaths per day in Istanbul during the summer months from 2004 to 2017 ranged from 101 to 187. The daily death rates were between 0.85 and 1.26 per hundred thousand residents. The average $T_i$ during the summer months in these years was between 22.3 °C and 25.1 °C. The descriptive characteristics of deaths and temperatures in the summer months each year are listed in Table 1.

The T95 of the daily mean temperatures in June, July, and August from 1971 to 2000 was 24.8 °C, 26.1 °C, and 26.2 °C, respectively. Based on the definition above, 30 heat waves were identified in Istanbul during the summer months between 2004 and 2017. The only year without any heat waves was 2004. Of the 30 heat waves, 20 (67%) had significantly higher crude death rates than during the reference period. A total of 4281 excess deaths were recorded during 20 heat waves that lasted 257 days in total. The start–end date and the length of each heat wave, the average $T_i$, maximum EHF, total EHF, number of excess deaths, rate ratios, and 95% confidence intervals are given in Table 2.

**Table 1.** Descriptive characteristics of deaths and temperatures in summer months by years.

| Years | Population | Daily Average Number of Deaths (Rate/100k) | Average of $T_i$s (°C) (Min–Max) | Number and Length (Days) of Heat Waves |
|---|---|---|---|---|
| 2004 | 11,910,733 | 101 (0.85) | 22.3 (16.8–26.9) | 0 (0) |
| 2005 | 12,128,577 | 106 (0.87) | 23.1 (15.9–27.4) | 1 (8) |
| 2006 | 12,351,506 | 112 (0.91) | 23.6 (16.5–28.7) | 2 (8, 10) |
| 2007 | 12,573,836 | 117 (0.93) | 25.0 (19.5–32.7) | 3 (10, 15, 7) |
| 2008 | 12,697,164 | 129 (1.02) | 24.2 (16.7–28.0) | 2 (7, 15) |
| 2009 | 12,915,158 | 134 (1.04) | 24.0 (19.8–27.4) | 1 (5) |
| 2010 | 13,255,685 | 150 (1.13) | 24.9 (17.4–30.1) | 2 (14, 25) |
| 2011 | 13,624,240 | 147 (1.08) | 23.6 (18.6–27.8) | 1 (16) |
| 2012 | 13,854,740 | 159 (1.15) | 25.1 (19.2–29.3) | 5 (5, 6, 12, 12, 9) |
| 2013 | 14,160,467 | 155 (1.09) | 24.2 (18.3–27.4) | 2 (5, 11) |
| 2014 | 14,377,018 | 173 (1.20) | 24.6 (18.5–28.4) | 4 (5, 5, 17, 8) |
| 2015 | 14,657,434 | 179 (1.22) | 24.3 (18.8–28.9) | 1 (29) |
| 2016 | 14,804,116 | 187 (1.26) | 25.0 (17.3–28.9) | 3 (18, 21, 7) |
| 2017 | 15,029,231 | 187 (1.24) | 24.3 (18.4–29.2) | 3 (7, 5, 12) |

Rate/100k, rate per hundred thousand; min, minimum; max, maximum.

**Table 2.** Characteristics of heat waves with a crude death rate higher than the reference period.

| HW | Start–End Date | Days | Average of Daily Mean Temp. (°C) | Max EHF | Total EHF | Number of Excess Deaths | RR | 95% CI | *p*-Value |
|---|---|---|---|---|---|---|---|---|---|
| 2005-1 | 30 Jul–6 Aug | 8 | 26.65 | 3.56 | 13.02 | 61 | 1.07 | 1.00–1.15 | 0.046 |
| 2006-1 | 25 Jun–2 Jul | 8 | 25.53 | 7.52 | 26.79 | 102 | 1.12 | 1.04–1.19 | 0.001 |
| 2006-2 | 14 Aug–23 Aug | 10 | 26.53 | 1.91 | 8.30 | 108 | 1.10 | 1.03–1.17 | 0.002 |
| 2007-1 | 21 Jun–30 Jun | 10 | 27.11 | 45.36 | 153.41 | 339 | 1.31 | 1.23–1.38 | <0.001 |
| 2007-2 | 17 Jul–31 Jul | 15 | 26.81 | 10.07 | 31.05 | 158 | 1.10 | 1.04–1.15 | 0.001 |
| 2007-3 | 20 Aug–26 Aug | 7 | 26.93 | 2.61 | 7.09 | 68 | 1.09 | 1.01–1.17 | 0.021 |
| 2008-1 | 23 Jun–29 Jun | 7 | 25.39 | 6.27 | 18.42 | 115 | 1.13 | 1.06–1.21 | <0.001 |
| 2008-2 | 12 Aug–26 Aug | 15 | 26.72 | 3.12 | 14.72 | 185 | 1.10 | 1.05–1.15 | <0.001 |
| 2010-1 | 14 Jul–27 Jul | 14 | 26.49 | 3.50 | 19.29 | 203 | 1.10 | 1.05–1.16 | <0.001 |
| 2010-2 | 29 Jul–22 Aug | 25 | 28.16 | 8.71 | 112.14 | 783 | 1.22 | 1.18–1.27 | <0.001 |
| 2011-1 | 16 Jul–31 Jul | 16 | 26.56 | 3.78 | 24.26 | 154 | 1.07 | 1.02–1.11 | 0.004 |
| 2012-1 | 11 Jun–15 Jun | 5 | 24.98 | 5.47 | 9.82 | 67 | 1.09 | 1.01–1.17 | 0.022 |
| 2012-3 | 7 Jul–18 Jul | 12 | 26.61 | 4.18 | 16.93 | 227 | 1.12 | 1.07–1.18 | <0.001 |
| 2012-4 | 21 Jul–1 Aug | 12 | 27.23 | 6.46 | 33.47 | 133 | 1.07 | 1.02–1.13 | 0.006 |
| 2013-2 | 12 Aug–22 Aug | 11 | 26.44 | 1.69 | 3.75 | 154 | 1.09 | 1.04–1.15 | <0.001 |
| 2015-1 | 24 Jun–28 Jun | 29 | 27.06 | 9.06 | 60.22 | 506 | 1.10 | 1.07–1.14 | <0.001 |
| 2016-1 | 16 Jun–3 Jul | 18 | 25.86 | 9.88 | 70.06 | 295 | 1.09 | 1.05–1.14 | <0.001 |
| 2016-2 | 23 Jul–12 Aug | 21 | 27.12 | 5.79 | 44.31 | 309 | 1.08 | 1.04–1.12 | <0.001 |
| 2016-3 | 17 Aug–23 Aug | 7 | 26.81 | 1.10 | 3.92 | 94 | 1.08 | 1.02–1.14 | 0.013 |
| 2017-1 | 27 Jun–3 Jul | 7 | 27.00 | 21.59 | 70.64 | 220 | 1.17 | 1.11–1.23 | <0.001 |
| Total | | 257 | | | | 4,281 | | | |

HW, heat wave; temp, temperature; max, maximum; EHF, Excess Heat Factor; RR, rate ratio; CI: confidence interval.

The 25th, 50th, 75th, and 95th percentiles of the EHF were calculated using the EHF values of all heat waves. The percentile for each day of a heat wave was presented in Figure 2.

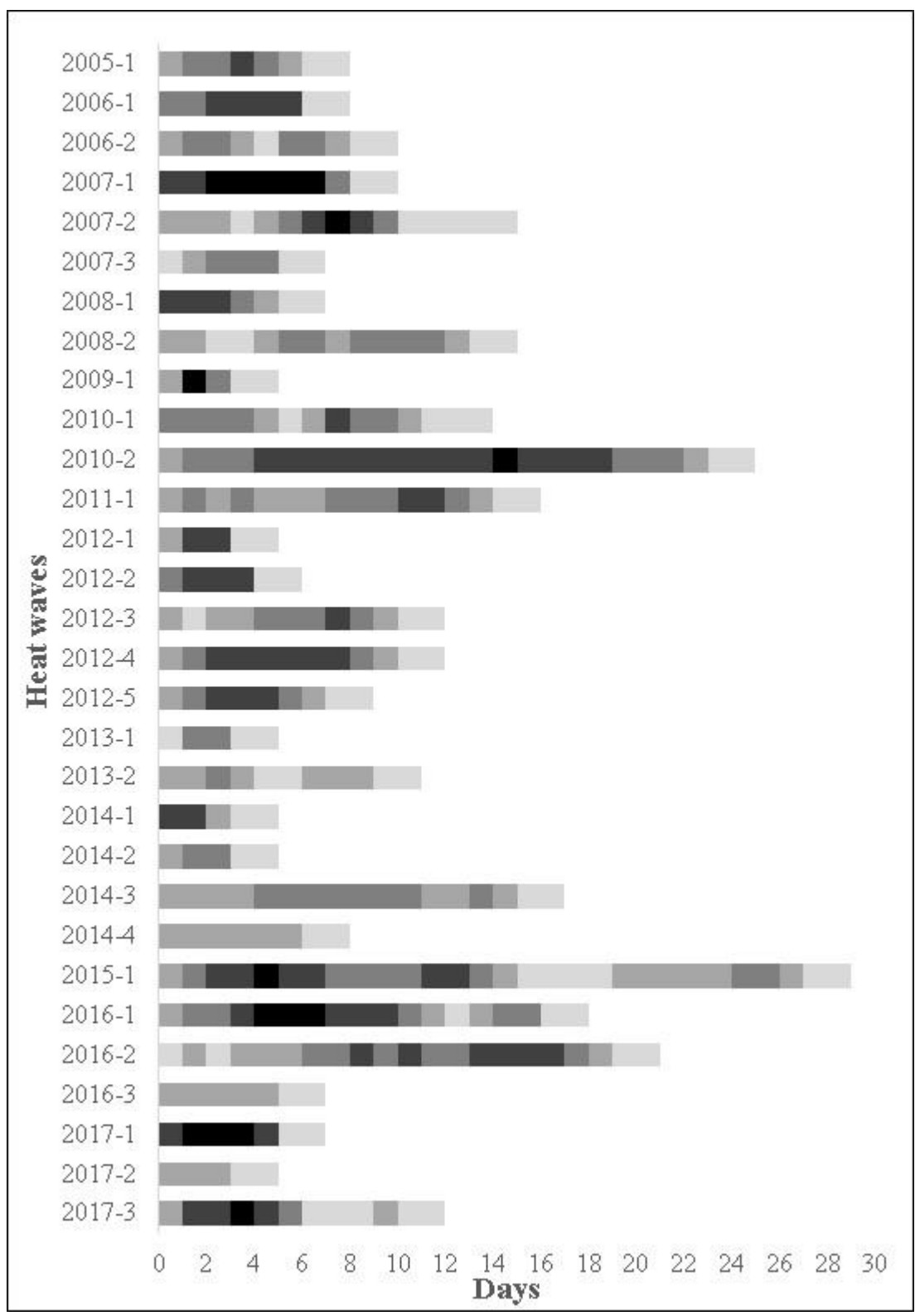

**Figure 2.** Percentiles of Excess Heat Factor (EHF) values for each day in heat waves (color scale from light to dark: <25th percentile, 2–50th percentile, 50–75th percentile, 75–95th percentile, ≥95th percentile).

The first heat wave in 2007 (2007-1), the second heat wave in 2010 (2010-2), and the first heat wave in 2017 (2017-1) had the highest rate ratios. In 2007-1, the heat wave lasted 10 days; the rate ratio was 1.31 (95% CI 1.23–1.38) and significantly increased ($p < 0.001$). This ratio corresponded with 339 excess deaths during 2007-1. In 2010-2, the heat wave lasted 25 days; the rate ratio was 1.22 (95% CI 1.18–1.27) and significantly increased ($p < 0.001$). This ratio corresponded with 783 excess deaths during 2010-2. In 2017-1, the heat wave lasted seven days; the rate ratio was 1.17 (95% CI 1.11–1.23) and significantly increased ($p < 0.001$). This ratio corresponded with 220 excess deaths during 2017-1. A total of 1342 excess deaths occurred during these three heat waves, which lasted 42 days in total. The number of daily excess deaths with EHF and EHI$_{sig}$ are given in Figures 3–5.

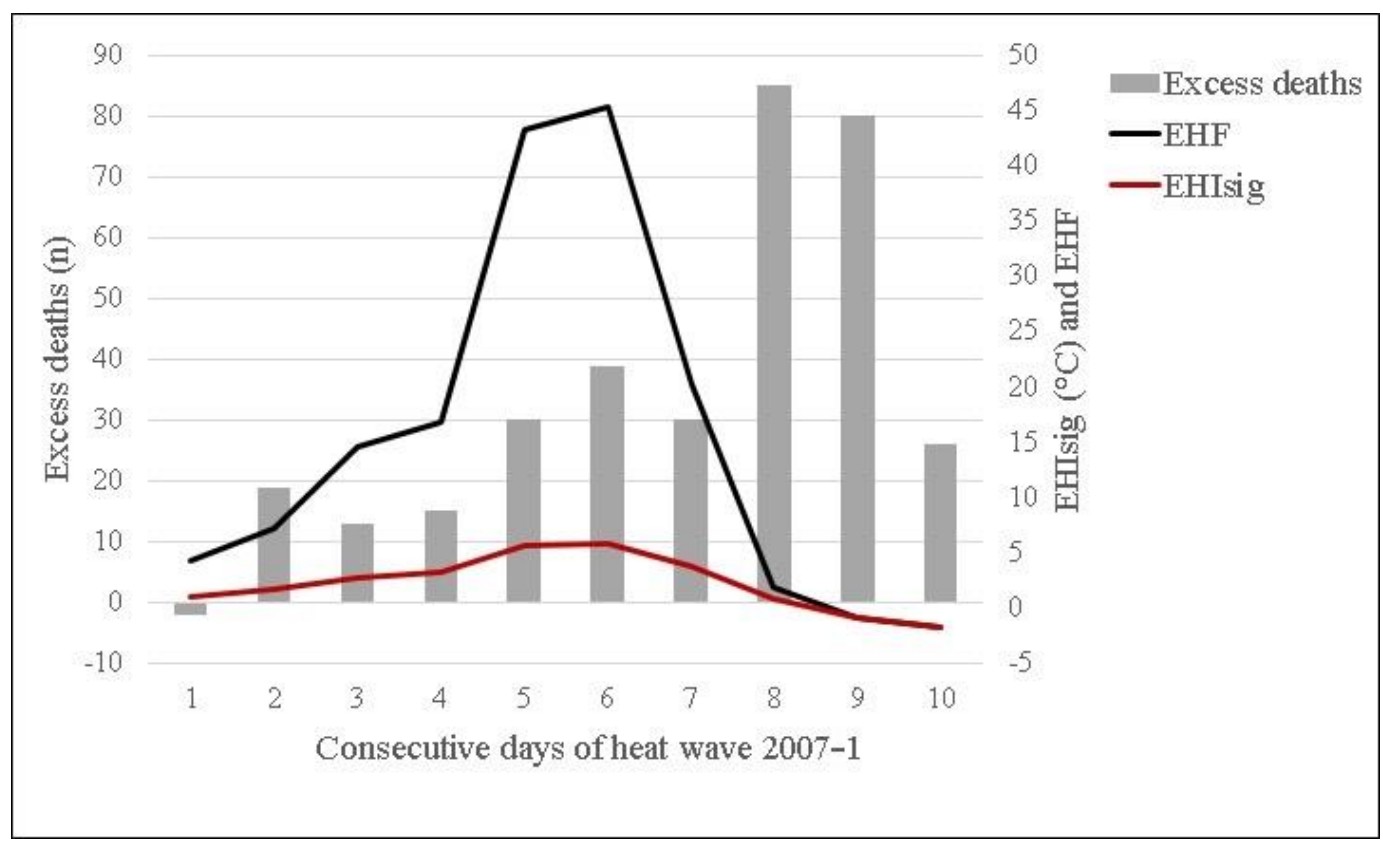

**Figure 3.** Number of daily excess deaths with Excess Heat Factor (EHF) and significance index (EHI$_{sig}$) values in the first heat wave of 2007.

Although the maximum EHF was similar to 2010-2 and total EHF to 2017-1, the increase in the rate ratio was less in the second heat wave of 2007 (2007-2) and the first heat wave of 2015 and 2016 (2015-1 and 2016-1). In 2007-2, which lasted 15 days, the rate ratio was 1.10 (95% CI 1.04–1.15) and significantly increased ($p = 0.001$). This ratio corresponded with 158 excess deaths during 2007-2. In 2015-1, which lasted 29 days, the rate ratio was 1.10 (95% CI 1.07–1.14) and significantly increased ($p < 0.001$). This ratio corresponded with 506 excess deaths during 2015-1. In 2016-1, which lasted 18 days, the rate ratio was 1.09 (95% CI 1.05–1.14) and significantly increased ($p < 0.001$). This ratio corresponded with 506 excess deaths during 2016-1.

None of the heat waves in the years 2009 and 2014 had significant increases in rate ratios ($p < 0.05$). There was also no significant increase in the risk of mortality in some of the heat waves that occurred in the following weeks of the previous heat wave, such as the second and fifth heat waves of 2012 (2012-2 and 2012-5) and the second and third heat waves of 2017 (2017-2 and 2017-3) (Table 3).

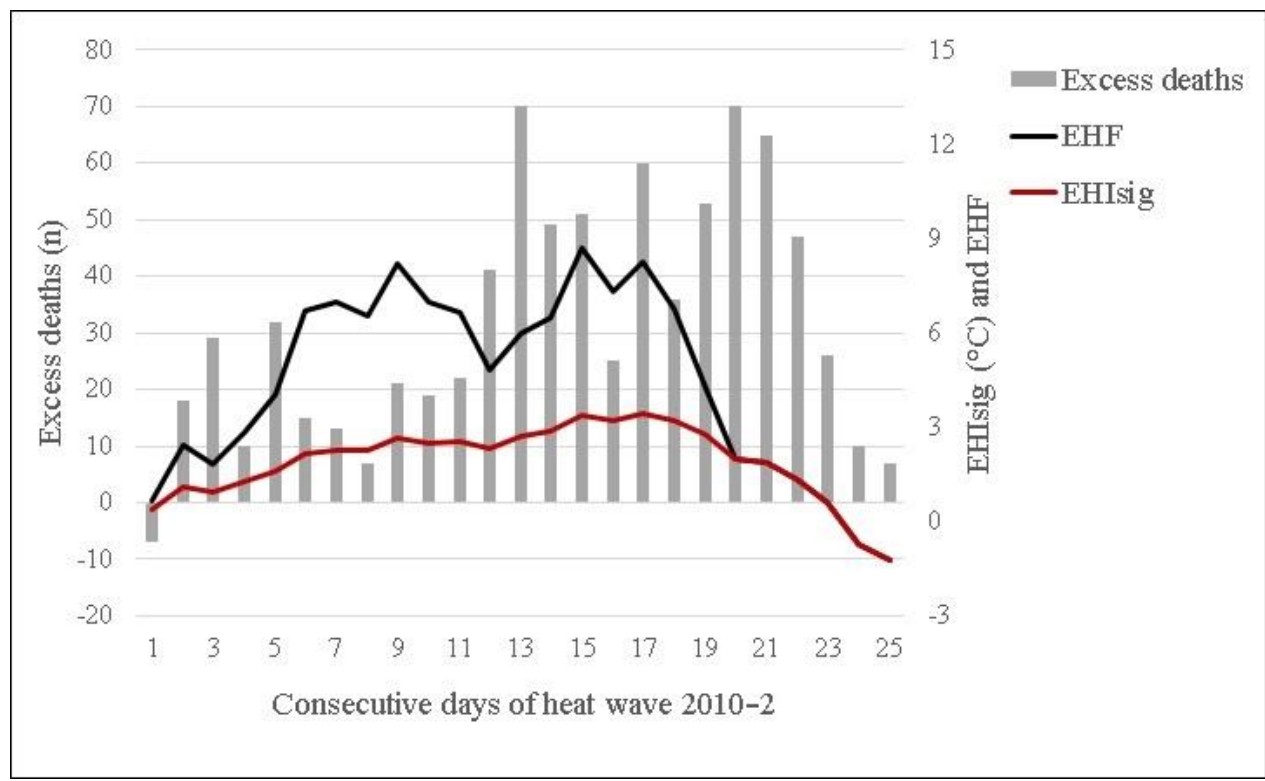

**Figure 4.** Number of daily excess deaths with Excess Heat Factor (EHF) and significance index (EHI$_{sig)}$) values in the second heat wave of 2010.

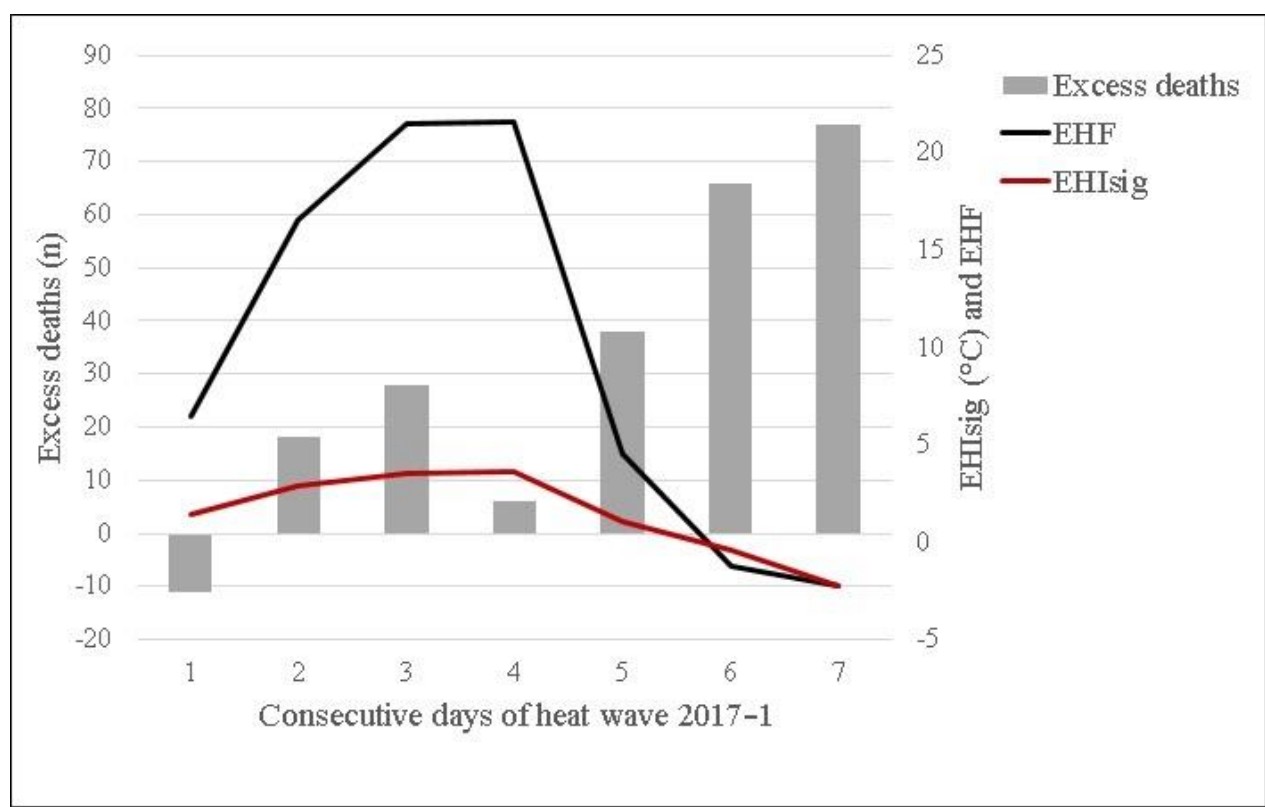

**Figure 5.** Number of daily excess deaths with Excess Heat Factor (EHF) and significance index (EHI$_{sig)}$) values in the first heat wave of 2017.

**Table 3.** Characteristics of heat waves with no significant increase in rate ratio.

| HW | Start–End Date | Days | Average of Daily Mean Temp. (°C) | Max EHF | Total EHF | RR | 95% CI | *p*-Value |
|---|---|---|---|---|---|---|---|---|
| 2009-1 | 21 Jun–25 Jun | 5 | 24.97 | 9.94 | 13.03 | 1.06 | 0.98–1.14 | 0.134 |
| 2012-2 | 21 Jun–26 Jun | 6 | 25.72 | 7.21 | 20.84 | 1.02 | 0.95–1.09 | 0.583 |
| 2012-5 | 3 Aug–11 Aug | 9 | 27.49 | 6.10 | 18.94 | 0.99 | 0.94–1.05 | 0.774 |
| 2013-1 | 23 Jun–27 Jun | 5 | 24.87 | 1.57 | 3.21 | 1.03 | 0.96–1.10 | 0.468 |
| 2014-1 | 24 Jun–28 Jun | 5 | 24.77 | 5.02 | 8.60 | 1.04 | 0.97–1.11 | 0.263 |
| 2014-2 | 9 Jul–13 Jul | 5 | 26.12 | 2.83 | 5.57 | 1.02 | 0.96–1.10 | 0.488 |
| 2014-3 | 21 Jul–6 Aug | 17 | 26.70 | 3.08 | 19.25 | 1.03 | 0.99–1.08 | 0.125 |
| 2014-4 | 10 Aug–17 Aug | 8 | 26.90 | 0.82 | 4.24 | 1.05 | 1.00–1.11 | 0.061 |
| 2017-2 | 23 Jul–27 Jul | 5 | 26.05 | 0.37 | 0.78 | 1.05 | 0.98–1.12 | 0.180 |
| 2017-3 | 2 Aug–13 Aug | 12 | 27.02 | 9.18 | 28.76 | 0.99 | 0.95–1.04 | 0.758 |

HW, heat wave; temp, temperature; max, maximum; EHF, Excess Heat Factor; RR, rate ratio; CI: confidence interval.

## 4. Discussion

In this study, in which EHF was used to determine heat waves, 30 heat waves with different lengths and intensities were identified in June, July, and August between 2004 and 2017. The total duration of these heat waves was 334 days. In 67% (n = 20) of the heat waves, the death rate was significantly higher than during the reference period. Twenty heat waves that lasted a total of 257 days caused an increased risk of mortality ranging between 7% and 31% and corresponded to a total of 4281 deaths.

In the study, the heat waves with the highest mortality risk were 2007-1, with an increase of 31%; 2010-2, with an increase of 22%; and 2017-1, with an increase of 17%. In total, 1342 (339 + 783 + 220) deaths occurred during these three heat waves, which lasted for 42 days (10 + 25 + 7). The maximum total EHFs of these heat waves were 45.36–153.41 °C$^2$, 8.71–112.14 °C$^2$, and 21.59–70.64 °C$^2$, respectively. The EHF values were above the 95th percentile for half of 2007-1 and 2017-1, and above the 75th percentile for more than half of 2010-2. Previous studies also reported similar results regarding the intensities and lengths of the heat waves in similar regions, including Istanbul, in these years. In a study determining the heat waves that have occurred in Turkey since 1950, it is reported that the intensity of the heat waves peaked in the years 2007, 2010, and 2017 [29]. In another study examining Euro-Mediterranean heat waves, the summers of 2007 and 2010 were found to be extremely hot, and the heat waves in these years were classified as high impact [30]. Similar results in the literature were found in terms of the increase in mortality rates. In the study evaluating extreme heat waves that occurred in Istanbul between 2013 and 2017, it was found that a heat wave that lasted for six days between 29 June and 4 July 2017 increased the risk of death by 21% [16]. A study in South Australia using EHF reported that the risk of death increased between 5% and 28% in moderate heat waves, while it increased between 22% and 50% in high-intensity heat waves [31].

In the study, it was determined that there were no heat waves in 2004, and that there was no significant increase in the mortality rate during the 2009 and 2014 heat waves. The average daily mean temperature in all heat waves in 2009 and 2014 was only 0.02 °C to 0.66 °C higher than T$_{95}$ of the same month. Moreover, although the maximum EHF in the 2009 heat wave was 9.94 °C$^2$, the EHF during this five-day heat wave was only greater than the 95th percentile for one day. Similarly, in 2014, there were no heat wave days with an EHF higher than the 95th percentile. In previous studies examining heat waves in Turkey and the Eastern Mediterranean, similar results were found for the temperatures in the years 2004, 2009, and 2014. In these studies, while no heat wave was identified in 2004 [4,13], it is seen that the intensity of heat waves was low in 2009 and 2014 [29]. Elevated EHI$_{accl}$ and

thus EHF are indicative of increased heat stress [25]. The low intensity of heat waves in 2009 and 2014 may have led to low heat stress and therefore no increase in mortality risk. However, this finding does not mean that low-intensity heat waves do not cause adverse health effects. In the evaluation of the health impacts of low-intensity heat waves, it may be more appropriate to use morbidity parameters, such as ambulance calls and hospital admissions, rather than mortality. In a study conducted in south Australia, it was found that ambulance calls and hospital admissions increased significantly during low-intensity heat waves, but there was no significant increase in deaths rates [31]. In another study that used EHF, it was found that the rate of admission to emergency services was related to the severity of the heat wave, and the rate of admission was higher in severe heat waves [32].

The intensity of the heat wave is an important determinant of the increased risk of death. However, not every day during a heat wave has the same intensity [27]. The pattern of temperature rise is also important when assessing the mortality risk. Adaptation mechanisms may not work adequately during sudden and intense temperature increases [33]. On the contrary, the gradual increase in temperatures may allow for adaptation [25]. Although their total EHF is similar to 2017-1, the lower risk of mortality in 2007-2, 2015-1, and 2016-1 may be due to the more sudden and intense temperature rise in 2017-1. While the EHF value was above the 95th percentile on the second day of 2017-1, the EHF values in 2007-2, 2015-1, and 2016-1 were not above the 95th percentile until the fifth through eighth days. Although the maximum EHF and time to reach maximum EHF were similar to 2010-2, the lower increase in the risk of death in these three heat waves may be due to the persistence of high EHF values for an extended period of time in 2010-2. Similar to our study, studies in the literature have also reported that long-term high temperatures have a greater effect on mortality [16,34].

Some studies indicate that there is a decrease in death rates, especially in the following weeks after extreme heat waves [35,36]. This short-term shift in mortality is called the harvesting effect. In particular, the elderly and people with chronic diseases are under increased risk from the adverse health effects of heat waves. According to the harvesting effect, heat waves particularly affect "people whose health is already so deteriorated as to die in the short term." Therefore, the death rate decreases in the weeks following the heat wave [36]. In the present study, it was observed that there was no significant increase in the mortality risk during the following wave in some successive heat waves. The lack of a significant increase in mortality rates in the 2012-2, 2012-5, 2017-2, and 2017-3 heat waves may be due to the harvesting effect.

Epidemiological studies have revealed that environmental events such as heat waves and air pollution have delayed health effects. Studies on air pollution report an increased mortality caused by air pollution several weeks after exposure to pollution [37]. However, this lag period is shorter in studies examining mortality due to heat waves [38]. In the present study, the maximum EHF and maximum excess deaths did not occur on the same day, as seen in Figure 3–5, which could be interpreted as a delayed health effect. On the other hand, in this study, the lag period was not added while determining the excess deaths because the lag period is already present in the heat wave definition determined by the EHF [24]. The two days ($i + 1$, $i + 2$) following the last day with positive EHF ($i$) are added to the heat wave even if the average temperature of those days ($T_{i+1}$ and $T_{i+2}$) is less than T95.

This study has several strengths. Unlike many other heat wave indicators, EHF also takes into account the adaptation (acclimatization) of individuals to temperature changes in the local climate throughout the year. Therefore, it may be more accurate than other definitions when examining the health effects of heat waves [24,32]. EHF is an indicator included in the guidelines for establishing an early warning system published by the WHO and WMO [25]. Moreover, it is already used by the Australian Bureau of Meteorology to forecast heat waves [24]. Higher values of EHF at high and sudden temperature increases may demonstrate a strong dose–response relationship between EHF and the health outcomes of heat waves [25,32].

There are some limitations of the study. Although mortality data in Istanbul have been collected and recorded for a long time by health authorities and local governments, the systematic classification of the daily number of deaths according to basic demographic characteristics and causes of death became possible only after the Death Notification System, which began in 2013. Therefore, only the increase in crude death rates during heat waves was evaluated in this study, and no evaluation was performed regarding cause-, age-, or sex-specific death rates. Since the mortality data used in the study did not include the place of death, the effect of urban heat islands on death rates could not be evaluated in the study. Since the data used in the study did not include the effects of socioeconomic level, housing quality, and similar variables that could cause variances in the health impacts, they could not be evaluated. Such data can enable better estimations of the health risks caused by heat waves and assist in the preparation of more precise action plans.

**5. Conclusions**

The present study, which used EHF to identify heat waves, showed that a total of 4281 excess deaths occurred in Istanbul during the twenty heat waves that lasted for a total of 257 days in the summer months of 2004–2017. It was also found that the increase in mortality risk was higher in heat waves where high EHF values were long lasting or increased rapidly. These results suggest that EHF is an indicator that can be used in establishing an early warning system in Istanbul. However, studies assessing the relationship of EHF with morbidity, such as ambulance calls, emergency service admissions, and hospitalizations, are also needed for Istanbul.

An early warning system alone may not be sufficient for preparing a heat-health action plan. In particular, the health system needs to be developed in this regard. Raising the awareness of health service providers regarding the impacts of heat waves which are increasing due to climate change and implementing policies that encourage collaboration with other industries can assist in developing the health system. Urban planning to prevent urban heat islands, ensure water and food security, and promote public awareness should be considered in order to prevent the negative effects of heat waves on health.

**Author Contributions:** Conceptualization, H.C.Ç., Ü.Ş. and G.C.; methodology, H.C.Ç. and G.C.; software, H.C.Ç.; validation, H.C.Ç. and G.C.; formal analysis, H.C.Ç.; data curation, H.C.Ç. and Ü.Ş.; writing—original draft preparation, H.C.Ç.; writing—review and editing, Ü.Ş. and G.C.; visualization, H.C.Ç.; supervision, Ü.Ş. and G.C. All authors have read and agreed to the published version of the manuscript.

**Funding:** The authors report that no funding has been received for the study.

**Institutional Review Board Statement:** The study was approved by the Clinical Research Ethics Committee of Istanbul University-Cerrahpasa (approval number: 43633, approval date: 3 March 2021).

**Informed Consent Statement:** Not applicable.

**Data Availability Statement:** The datasets analyzed during the current study are available from the corresponding author on reasonable request.

**Acknowledgments:** The authors would like to thank Ayhan Koç and Istanbul Cemeteries Department for the mortality data used in the study.

**Conflicts of Interest:** The authors declare that there is no conflict of interest regarding the publication of this paper.

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
