# Peer review of "A Step to Develop Heat-Health Action Plan: Assessing Heat Waves’ Impacts on Mortality"

_atmosphere, doi:10.3390/atmos13122126_

Round 1

Reviewer 1 Report

Dear researchers,

Your work points to a current and vital public health issue. Although it is an ecological study, it will guide countries with limited resources who want to establish early warning systems for heat waves and create a hot health action plan. Please find attached a few changes that I have suggested by highlighting them on your article. Apart from these, this work can be published.

It is possible to improve the study by using statistical analysis methods considering the time factor and various confounding factors (air pollution etc.). I hope you continue your work on the subject.

Reviewer 2 Report

This work aims to evaluate excess deaths during heat waves between the 13 summer months of 2004 and 2017 in Istanbul and to determine a definition of heat waves that can 14 be used in developing an early warning system, a part of prospective urban heat-health 15 action plans.

The paper is interesting, well-organized, and well-written. Data, methods, results, and limitations are precise.

I present two key points only:

1. Quality of maps, figures, and tables:

 Figure 1 - Authors must provide a map with scale, orientation indicators, and source notes.

 Figures 2 to 5 - Authors must improve the quality of those Figures.

 Tables 1 to 3 - What is "n"? "Number of"? Please, be clear. 

2. A conceptual point: why is this paper related to an early warning system and not strictly to a detection one?

Reviewer 3 Report

See attachment

Reviewer 4 Report

Review report for Manuscript ID: atmosphere-2050670

Title: A Step to Develop Heat-Health Action Plan: Assessing Heat Waves’

Impacts on Mortality

Authors: Hazal Cansu Culpan, Umit Sahin, Gunay Can

General comments:

The study 'A Step to Develop Heat-Health Action Plan: Assessing Heat Waves’ Impacts on Mortality' aims to evaluate excess deaths during heat waves between summer months of 2004 and 2017 in Istanbul and to determine a definition of heat waves that can be used in the development of an early warning system, a part of prospective urban heat-health action plans; heat waves were determined using the Excess Heat Factor, an index based on a three-day-averaged daily mean temperatures.

The topic of study is lately highly researched field and interesting for the wider population, the purpose of the work is clearly defined, the methodology is adequately presented. The work fit the journal scope (air quality, human health). The study lists 36 references, of which 11 are for the period of the last 5 years, all cited references are relevant regarding the topic, but there are some misquotes (wrongly cited source for claim). List does not include an excessive number of self-citations.

The study reported have been carried out in accordance with generally accepted ethical research standards and it was approved by the Clinical Research Ethics Committee (Approval number and date are given); it was reported according to Strengthening the Reporting of Observational Studies in Epidemiology (STROBE) Statement.

The experimental design is appropriate to test the hypothesis, death rate during heat waves and non-heat wave days during summer months were compared with a Z test (p<0.05); all needed details (and equations) are given in the methods section. Results are interpreted appropriately and are significant; conclusions are justified and supported by the results; at the end, the limitations of this study are also explained.

Some figures could be more easier to interpret and understand; there are also some errors regarding interpretation of the data (wrong numbers in text regarding the data in Tables), certain abbreviations could be used, especially in Tables.

The language is appropriate and understandable.

Specific comments 

Line 18: redundant word ‘were’ (repeated twice)

Line 19: you should put the value for statistical significance in ‘Methods’, not in Abstract

Line 23: here is 'Extreme Heat Factor', above you speak about 'Excess heat Factor'

Lines 43-44: you have 'In the 2010 heat wave in India, more than 1,300 excess deaths were reported in Ahmedabad alone [8]' – but Reference n.8 (Semenza et al., 1996) is about something else (Heat-related deaths during the July 1995 heat wave in Chicago)

Lines 44-45: you have 'Extreme heat waves in 2017 in Pakistan caused one thousand excess deaths [9]' – but Reference n.9 (Azhar et al., 2014) is about something else (Heat-related mortality in India: excess all-cause mortality associated with the 2010 Ahmedabad heat wave).

Line 91: this is the first time you have used an abbreviation (Ti) for Daily mean temperatures; use it further on in the text and in the headers of the Tables 1,2,3

Lines 91-95: according to the text - is it in fact used a two-day temperature average and not a daily one?

Line 164: you have 'Daily death rates were between 0.85 and 1.24…', but in Table 1 there is a rate 1.26 for year 2016

Figure 3: shows 11 days of the first heat wave of 2007 (x axis), but in Lines 191-192 you have ‘In 2007-1, the heat wave lasted 10 days'

Figures 3,4,5:

-put the legend for the columns and lines inside the figure to make it clearer;

-put the name of the variable on the secondary y axis, not just the units

-label the x-axis more clearly, for example like ‘consecutive days of the heat wave and the wave label (e.g. 2007-1)…’

Line 216: you have 'In 2015-1, which lasted 19 days,' – but in Table 2 you indicate a length of 29 days for the same period

Table 2 and 3:

-in Table 2 you use the notation 1.00-1.15 for the 95% CI, but in Table 3 it is different (1.00, 1.15), it should be uniform;

-Table 2, first line for CI, should also have only two decimal places as others, not three

-why the explained abbreviation HW for heat wave in the bottom line if it is not in the table

Line 262: you have 'it is seen that the intensity of heat waves was low in 2009 and 2014 [4,13,28].' – but Reference n.4 (Kuglitsch 2010) quantify Heat wave changes in the eastern Mediterranean between 1960 and 2006;  and Reference n.13 (Unal 2013) study Summer heat waves over western Turkey between 1965 and 2006 – so those 2 references are not an appropriate citations for years 2009 and 2014

Line 280: write 'SHD' with whole words or explain the abbreviation

Round 2

Reviewer 3 Report

1) I accept your response on the delayed mortality effect.

2) I cannot accept your response on the method description: a) Generally reference(s) is needed when we do not use a very widely used method; b) 'var' normally means variance, but not in your formulas; c) I can say only that trying to use Eq. (6) it does not function as a hypothesis test statistic. Maybe I make error(s); but I suggest you not to insist on formulas 6&7.

I ask you:

1) Please, delete formulas (6) and (7);

2) Give a reference to the Z-test you apply (if you can).

Apart from this issue, only minor language check is needed before publication.
